# Clorfl86/RHEX Is a Negative Regulator of SCF/KIT Signaling in Human Skin Mast Cells

**DOI:** 10.3390/cells12091306

**Published:** 2023-05-03

**Authors:** Kristin Franke, Gürkan Bal, Zhuoran Li, Torsten Zuberbier, Magda Babina

**Affiliations:** 1Institute of Allergology, Charité—Universitätsmedizin Berlin, Corporate Member of Freie Universität Berlin and Humboldt-Universität zu Berlin, Hindenburgdamm 30, 12203 Berlin, Germany; 2Fraunhofer Institute for Translational Medicine and Pharmacology ITMP, Immunology and Allergology IA, Hindenburgdamm 30, 12203 Berlin, Germany

**Keywords:** mast cell, RHEX, C1orf186, survival, apoptosis, signal transduction, ERK1/2, p38, immediate-early genes, capicua, skin

## Abstract

Mast cells (MCs) are key effector cells in allergic and inflammatory diseases, and the SCF/KIT axis regulates most aspects of the cells’ biology. Using terminally differentiated skin MCs, we recently reported on proteome-wide phosphorylation changes initiated by KIT dimerization. C1orf186/RHEX was revealed as one of the proteins to become heavily phosphorylated. Its function in MCs is undefined and only some information is available for erythroblasts. Using public databases and our own data, we now report that RHEX exhibits highly restricted expression with a clear dominance in MCs. While expression is most pronounced in mature MCs, RHEX is also abundant in immature/transformed MC cell lines (HMC-1, LAD2), suggesting early expression with further increase during differentiation. Using RHEX-selective RNA interference, we reveal that RHEX unexpectedly acts as a negative regulator of SCF-supported skin MC survival. This finding is substantiated by RHEX’s interference with KIT signal transduction, whereby ERK1/2 and p38 both were more strongly activated when RHEX was attenuated. Comparing RHEX and capicua (a recently identified repressor) revealed that each protein preferentially suppresses other signaling modules elicited by KIT. Induction of immediate-early genes strictly requires ERK1/2 in SCF-triggered MCs; we now demonstrate that RHEX diminution translates to this downstream event, and thereby enhances NR4A2, JUNB, and EGR1 induction. Collectively, our study reveals RHEX as a repressor of KIT signaling and function in MCs. As an abundant and selective lineage marker, RHEX may have various roles in the lineage, and the provided framework will enable future work on its involvement in other crucial processes.

## 1. Introduction

Mast cells (MCs) are tissue-resident cells of hematopoietic origin that appear early in evolution and are strategically located at the interface of host and environment in tissues like the skin, gut, and respiratory tract [1,2,3,4].

As principal effector cells of IgE-mediated type-I and IgE-independent pseudo-allergic hypersensitivity reactions, MCs contribute to acute phenomena like food allergy and anaphylaxis, as well as to complex diseases like rhinitis, asthma, eczema, and urticaria [5,6,7,8,9,10,11,12,13,14,15,16,17,18,19,20,21].

Conversely, MCs also act as early sentinels sensing pathogens and perturbations of homeostasis, and, thereby, they aid in mounting proper immune responses [2,3,4,22,23]. In fact, their strategic position close to blood vessels, nerves, and other structures (e.g., hair follicles in the skin) predestines MCs to recognize environmental factors like allergens and venoms together with endogenous substances like cytokines, neuropeptides, and alarmins to integrate multiple signals and orchestrate subsequent responses [8,23,24].

The (c-)KIT/SCF (stem cell factor) axis is a major receptor tyrosine kinase (RTK) system in hematopoiesis [25,26,27,28]. It is expressed early on at the level of hematopoietic stem cells but is gradually downregulated as blood cells differentiate into their mature forms. In contrast, MCs retain high levels of KIT as fully differentiated cells [29], and KIT controls MC (precursor) expansion, differentiation, survival, and function throughout the cells’ lifespan [25,26,29]. KIT also plays major roles in other lineages like melanocytes and germ cells [27]. The binding of SCF to KIT leads to receptor dimerization and activates KIT’s intrinsic tyrosine kinase activity. This is followed by phosphorylation of key amino acid residues in the cytoplasmic region of KIT, followed by a series of phosphorylation events, including PI3K, MAPKs, and STAT3/5 [27,28]. The different modules are likely activated to variable degrees in distinct lineages. For skin MCs, we recently reported a dominant MEK/ERK module [30]. Since KIT regulates survival and proliferation, it can be oncogenic when aberrantly expressed and/or mutated. Therefore, somatic gain-of-function mutations (especially the D816V mutation) underlie systemic mastocytosis [27,29,31]. 

To provide refreshed in-depth insights into the SCF/KIT signaling network, we recently applied a global mass-spectrometry phosphoproteomics approach to SCF-stimulated skin MCs. Proteome-wide phosphorylation changes initiated by KIT dimerization were massive, with over 50% of the detectable phosphosites being regulated by SCF (over 5500 events in total) [30].

One strongly regulated novel KIT target turned out to be C1orf186/RHEX (regulator of human erythroid cell expansion) [32]. We found 8 phosphosites on the protein, of which 5 were significantly upregulated and one was downregulated by SCF (see also Figure 1f). Positive regulation sites included the two tyrosines Y-132 and Y-141 previously reported to be increased in erythroblastic cells in response to erythropoietin (EPO) [32]. These two residues also appear in a total of 4 and 7 high-throughput assays on the PhosphoSite Plus database, respectively (https://www.phosphosite.org/homeAction.action, last accessed on 27 February 2023). S-56 and S-128, also reported in the database, were likewise detected in MCs. In our effort, the site most vastly changing its phosphostatus was S-80, which is not yet listed on the PhosphoSite Plus database and may thus be MC-specific (log2 fold change ≈ 5) [30].

So far, nothing is known about RHEX’s role in MCs and little across the body. RHEX is conserved between humans and primates but is absent from mouse, rat, and lower vertebrates [32]. For the UT7epo cell line and primary CD34+-derived erythroblasts, EPO not only induced RHEX phosphorylation, but it was also disclosed to operate as a positive regulator of colony growth and signal transduction, including ERK1/2 phosphorylation [32]. RHEX’s substantial change in phosphorylation following SCF stimulation led us to address its functional implication in skin MCs. We report that, in contrast to its positive role in erythropoiesis, RHEX interferes with KIT signaling and survival promotion. Together with the recently uncovered capicua, RHEX is therefore revealed as another repressor of the KIT tyrosine kinase in MCs, though the two proteins exert suppression by distinct mechanisms. Since RHEX expression is also found in immature MC lines, RHEX’s role in various aspects of MC biology from early precursors to fully mature stages will be of substantial interest in the future.

## 2. Materials and Methods

### 2.1. Cells and Treatments

MCs were isolated from human foreskin tissue as described [33]. Each mast cell preparation/culture originated from several (2–12) donors to achieve sufficient cell numbers, as routinely performed in our lab [34,35,36,37,38]. Written, informed consent was obtained from the patients, or their legal guardians, and the study was approved by the university ethics committee (protocol code EA1/204/10, 9 March 2018). The experiments were conducted according to the Declaration of Helsinki Principles. Briefly, the skin was cut into strips and treated with dispase (26.5 mL per preparation, activity: 3.8 U/mL; Boehringer-Mannheim, Mannheim, Germany) at 4 °C overnight, the epidermis was removed, and the dermis finely chopped, and digested with 2.29 mg/mL collagenase (activity: 255 U/mg; Worthington, Lakewood, NJ, USA), 0.75 mg/mL hyaluronidase (activity: 1000 U/mg; Sigma, Deisenhofen, Germany), and DNase I at 10 µg/mL (Roche, Basel, Switzerland). Cells were filtered stepwise from the resulting suspension (using 100 and 40 µm strainers, Fisher Scientific, Berlin, Germany). MC purification finally was achieved by anti-human c-Kit microbeads (#130-091-332) and the Auto-MACS separation device (both from Miltenyi-Biotec, Bergisch Gladbach, Germany), giving rise to 98–100% pure preparations (FACS double-staining of KIT/FcεRI (anti-FcεRI eBiosciene #11-5899-42, Fisher Scientific; anti-CD117 Miltenyi-Biotec #130-111-593) and acidic toluidine blue (Sigma) staining, 0.1% *w*/*v* in 0.5 N HCl (Fisher Scientific), as described [39,40].

MCs were cultured in the presence of SCF, and IL-4 was freshly provided twice weekly when cultures were readjusted to 5 × 10^5^/mL. MCs were automatically counted by CASY-TTC (Innovatis/Casy Technology, Reutlingen, Germany) [34,41]. The leukemic MC line HMC-1 [42] was cultured at 5 × 10^5^–1 × 10^6^/mL using the same medium as for skin MC and fed three times a week. The LAD2 MC line, established from a patient with MC sarcoma [43], was kept in Stem-Pro 34 SFM (Fisher Scientific), supplemented with 100 U/mL Penicillin, 100 µg/mL Streptomycin, 2 mM L-glutamine, and 100 ng/mL SCF. Cells were cultured at 5 × 10^5^–1 × 10^6^/mL and were semi-depleted once a week.

### 2.2. Accell^®^-Mediated RNA Interference

A well-established and efficient siRNA method for skin MCs was used [30,35,37,44,45,46,47,48], exactly as described in our most recent work [49]. In brief, skin MCs were transfected twice (on day 0 and day 1) by RHEX-targeting siRNA, control siRNA, or CIC-targeting siRNA (further control) (each at 1 µM) for a total of 2 or 3 d in Accell^®^ medium (Dharmacon, Lafayette, CO, USA) (supplemented with Non-Essential Amino Acids and L-Glutamine (both from Carl Roth)). The “smart pools of 4” (Dharmacon) were used to target RHEX (E-033700-00-0050) and CIC (E-015185-00-0050). Transfections were performed in the presence of a low concentration of SCF (10 ng/mL) to maintain survival. SCF was provided either once (on day 0, when harvest was on day 2) or twice (on day 0, and day 1, when harvest was on day 3).

For immunoblot analysis and RT-qPCR, cells were stimulated with SCF (100 ng/mL) or PBS as a control for 25 min on day 2 and immediately frozen in RNA extraction buffer (for RT-qPCR). For immunoblot, SCF stimulation (100 ng/mL) was for 5, 10, or 15 min, respectively, as given in the figure legends; the procedures are further described under Section 2.3. For apoptosis measurements, cells were harvested after 3 d.

### 2.3. Immunoblot Analysis 

Appropriately pretreated MCs were collected by centrifugation and immediately solubilized in SDS-PAGE (Sodium Dodecyl Sulphate-Polyacrylamide Gel Electrophoresis) sample buffer and boiled for 15 min (whole-cell lysates). Samples of equal cell numbers were subjected to immunoblot analysis. Membrane blocking was performed in 5% (*w*/*v*) low-fat milk powder (Carl Roth, Karlsruhe, Germany) solution for 30 min. The following primary antibodies were purchased from Cell Signaling Technologies (Frankfurt am Main, Germany) and are as follows: anti-p-ERK1/2 (T202/Y204, #9101), anti-pp38 (T180/Y182, #9211), anti-p-STAT5 (Y694, #9359), anti-pAKT (S473, #9271), anti-β-Actin (#4967), anti-α-actinin (#6487), and anti-Cyclophilin B (#43603). The anti-RHEX (C1orf186, PA5-25856) antibody was purchased from Fisher Scientific.

As detection antibody, a goat anti-rabbit IgG peroxidase-conjugated antibody was used (Merck, Darmstadt, Germany, #AP132P). For consecutive development of several molecules on the same membrane, the antibodies (primary and secondary) were removed from the membrane after each detection step by incubation in 0.5 N NaOH (Carl Roth, Karlsruhe, Germany) for 15 min. After each stripping step, the membrane was blocked in 5% (*w*/*v*) low-fat milk powder for 30 min (as above), followed by incubation with the next primary antibody. Proteins were visualized by a chemiluminescence assay (Weststar Ultra 2.0, Cyanagen, Bologna, Italy), according to the manufacturer’s instructions. Bands were recorded on a chemiluminescence imager (Fusion FX7 Spectra, Vilber Lourmat, Eberhardzell, Germany). Semi-quantification of recorded signals was performed using the ImageJ software (Rasband, W.S., ImageJ, U.S. National Institutes of Health, Bethesda, MD, USA, https://imagej.nih.gov/ij/, 1997–2018, Software Version 1.48v/Java 1.6.0_20 (64-bit) (last accessed on 12 December 2022)). Individual intensity values for the detected proteins were normalized to the intensity of the housekeeping proteins cyclophilin B, α-actinin, and/or β-actin of the same membrane.

### 2.4. YoPro-1/Propidium Iodide (PI) Staining

Membrane permeability/apoptosis were determined with the YoPro-1/PI method, as described [30,37,45,50]. Briefly, cells were stained with the YoPro-1^TM^ dye (Fisher Scientific) and PI (BD Biosciences, Heidelberg, Germany) for 25 min on ice. Stained cells were measured on the MACSQuant^®^ Analyzer10 (Miltenyi Biotec, Bergisch-Gladbach, Germany) or the BD Calibur (Becton Dickinson, Heidelberg, Germany). Cells were not gated on a particular subpopulation (by FSC/SSC profile), but the proportions of viable cells were determined on the entire population. Data were analyzed with the FlowJo analysis software (FlowJo LLC, Ashland, OR, USA). YoPro^TM−1^ staining can detect both early and late apoptotic cells (the latter also positive for PI). The clearly double-negative cells were considered viable, and used for calculations.

### 2.5. Caspase-3 Activity

Caspase-3 activity of MCs was detected using a luminometric assay kit (Caspase-Glo 3/7; Promega, Mannheim, Germany), according to the manufacturer’s instructions and as previously described [37]. The assay provides a proluminescent caspase-3/7 substrate, which contains the sequence DEVD that is cleaved to release luminescence. Luminescence detection was performed using the VICTOR X5 2030 Multilabel HTS Microplate Reader (Perkin Elmer, Berlin, Germany) operated with the standard luminescence protocol. 

### 2.6. Reverse Transcription-Quantitative PCR (RT-qPCR)

MCs (at 5 × 10^5^ cells/mL) were stimulated with SCF (100 ng/mL) for 25 min, after which time cells were harvested for RNA extraction. Briefly, RNA was isolated using the NucleoSpin RNA kit from Machery-Nagel (Düren, Germany), following the manufacturer’s instructions. cDNA synthesis (reverse transcription kit from Fisher Scientific) and RT-qPCR were performed using optimized conditions as described elsewhere [33], using materials from Roche (Roche Diagnostics, Mannheim, Germany). The primer pairs are summarized in Table 1. They were synthesized by TibMolBiol, Berlin, Germany. The 2^−ΔΔCT^ method was used to quantify the expression levels of the target genes relative to three reference genes (appearing at the end of Table 1).

### 2.7. Statistics

Statistical analyses were carried out using PRISM 8.0 (GraphPad Software, La Jolla, CA, USA). Comparisons between two groups were performed using the paired Student’s *t*-test (when data were normally distributed), or Wilcoxon test (not normally distributed). A *p* value of less than 0.05 was considered statistically significant. For comparisons across more than two groups, an ordinary one-way ANOVA with Tukey’s multiple comparison test was used.

## 3. Results

### 3.1. RHEX Is Highly Expressed in MCs

Analyzing C1orf186/RHEX expression across the body in publicly available datasets revealed a highly restricted expression pattern. In the comprehensive atlas from the FANTOM5 consortium [51,52], abundant expression (≈1000 tpm (transcripts per million)) was detected only in MCs (isolated from human skin), while the mean of all non-MCs in the atlas was below 1 tpm (Figure 1a). Expression remained largely constant in MCs stimulated by FcεRI crosslinking (Figure 1a). Several myeloid leukemia cells were positive, but expression was much lower than in MCs. CD34+ cells differentiated along the erythrocytic pathway expressed higher levels than hematopoietic stem and progenitor cells (HPSCs), supporting the previously documented role in erythropoiesis [32] (Figure 1a). A recent skin-focused atlas [53] likewise demonstrated that RHEX was below detection in almost all cells, while expression was concentrated in skin MCs, followed by some dendritic cell subsets (Figure 1b). A larger version of the upper part of Figure 1b is presented as Appendix A.

We inquired whether RHEX expression is a general hallmark of the lineage, i.e., also found in intermediately differentiated and immature MC lines [42,54,55]. RHEX mRNA expression was therefore quantified in skin MCs and the mast cell lines HMC-1, and LAD2 side-by-side (cell lines kindly provided by Drs Butterfield and Kirshenbaum, respectively). With ct values of 19–24, i.e., in the same range as the highly expressed housekeeping gene PPIB (cyclophilin B), RHEX was abundantly expressed across MC types (Figure 1c). However, some differences were detectable, and the most immature MCs expressed the gene at a lower level, i.e., in the rank order of HMC-1 < LAD2 < skin MCs. In addition, the more mature 5C6 subclone of the HMC-1 cell line [56] showed greater expression than the parental cell line (data not shown). Even though expression was higher in the most mature (untransformed) MC type, expression in HMC-1 and LAD2 cells can be regarded as robust considering the very limited expression pattern in the body (Figure 1a) and the skin (Figure 1b).

We employed publicly available datasets to explore whether RHEX is also expressed by human cord-blood-derived MCs (hCBMCs). By RNA-seq, Dwyer et al. found robust expression of RHEX with a mean of 874 tpm (calculated by GEO2R-DESeq2 using their original datasets) across four samples (unstimulated and IL-4-treated). For comparison, the expression of FCER1A (encoding the α-chain of the high-affinity IgE receptor, one of the most extensively studied markers of the MC lineage) averaged 305 tpm [57]. IL-4 had no significant effect on RHEX expression (while it upregulated FCER1A) [57]. Higher transcript count for RHEX vis-à-vis FCER1A was confirmed in yet another study employing hCBMCs, whereby RHEX expression averaged 7.5 logCPM (log counts per million) (derived from Table E2 and Table E7 of the original publication), as opposed to FCER1A at 4.9 logCPM [58]. These data further underline that high RHEX mRNA expression is a universal hallmark of human MCs.

At the protein level, RHEX expression could be easily detected by immunoblot analysis in skin MCs and MC lines, whereby the same rank order of HMC-1 < LAD2 < skin MCs was confirmed (on comparison to three housekeeping proteins) (Figure 1d). RHEX gave a single band at ≈25 kDa in accordance with its previously reported size in erythroblasts [32]. In the publicly available MC proteome published by Plum et al. [59], considerable expression was found in skin and adipose-derived MCs by IBAQ (intensity-based absolute quantification) score, while PBMCs (peripheral blood mononuclear cells) were negative (Figure 1e). In their datasets, RHEX protein enrichment in skin MCs over PBMCs was among the top 100 most enriched proteins (Log2 FC ≈ 4.4) [59]. As mentioned in the Introduction section, RHEX was also striking because of its considerable change in phosphorylation upon SCF activation [30]. The affected amino acid residues detected in MCs are depicted in Figure 1f. Overall, changes were more pronounced after 8 vis-à-vis 30 min, suggesting that RHEX acts as an early responder.

Therefore, by all available information, RHEX is highly enriched in MCs and can be regarded as a novel lineage characteristic.

**Figure 1 cells-12-01306-f001:**
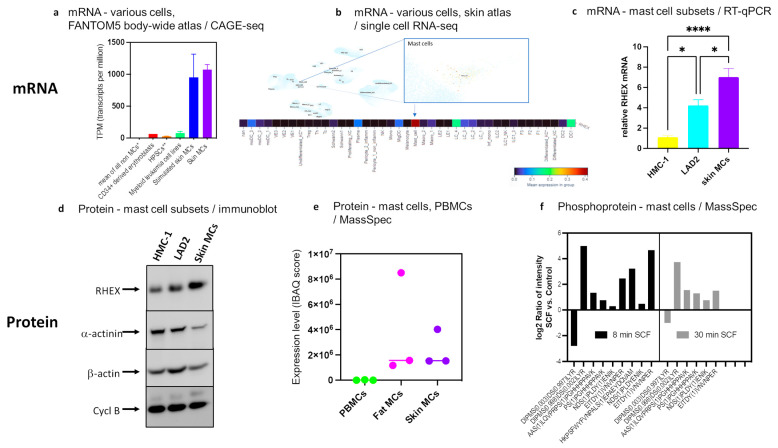
RHEX is a new lineage marker of MCs: (**a**) Data from the comprehensive FANTOM5 atlas [51,52] indicate highly enriched expression in MCs. The atlas is freely accessible via the FANTOM Zenbu genome browser (https://fantom.gsc.riken.jp/zenbu/gLyphs/#config=uPk5-KZcGxzk_88_pJlCc;loc=hg19::chr1:206226531..206300577+, last accessed on 27 February 2023); * mean expression across all FANTOM5 samples (1829 libraries) except for MCs (9 libraries); ** Hematopoietic stem and progenitor cells (HSPCs): Mean RHEX expression in CD34+ and CD133+ cells from bone marrow and blood; stimulated skin MCs: skin MCs that were stimulated with an activating anti-FcεRIα antibody for 2.5–16 h, as described in detail in the corresponding publication [60]. (**b**) Expression of RHEX mRNA across the skin atlas, which is accessible here: https://cells.ucsc.edu/?ds=healthy-human-skin&gene=RHEX, last accessed on 27 February 2023. Upper part: Clusters defining different cell types identified in the skin; RHEX expression is indicated by brown staining (very low across the atlas). A magnified section that zooms in on MCs is given on the right. A larger version of this figure is presented as Appendix A. Lower part: The different cell types across the atlas have been color-coded according to the key given below with red marking high, purple low expression. (**c**) Relative expression of RHEX mRNA across MC subsets by RT-qPCR. RHEX expression was normalized to three housekeeping genes (ACTB, HPRT, and PPIB, the latter encoding cyclophilin B), and mean expression in HMC-1 cells was set as 1. n = 8 for HMC-1, 6 for LAD2, and 12 for skin MCs. * *p* < 0.05; **** *p* < 0.0001. (**d**) RHEX protein expression in the three MC subsets by immunoblot. Different proteins with housekeeping-like properties were visualized consecutively on the same membrane to monitor protein loading (cycl B = cyclophilin B). (**e**) RHEX expression across the MC proteome on comparison to PBMCs [59]. IBAQ = intensity-based absolute quantification, PBMCs = peripheral blood mononuclear cells. The bar shows the median. (**f**) RHEX phosphosites and their change upon SCF stimulation after 8 and 30 min in skin MCs according to our recent phosphoproteome for the SCF/KIT axis [30]. For the experiment shown, MCs were deprived of growth factors overnight and stimulated with SCF at 100 ng/mL for 8 and 30 min, respectively, as described in greater detail in [30]. The identified amino acids are given on the x-axis.

### 3.2. RHEX Attenuation Supports MC Survival

To study RHEX function, we applied our RNA interference protocol successfully used to reduce multiple proteins in skin MCs [30,35,45,46,47,49]. The knockdown efficiency was ≈52% (i.e., down to 48%, Figure 2a) and thus within the range found for other targets, including MRGPRX2, CIC, STAT5, JNK (MAPK8), p38 (MAPK14), ERK1 (MAPK3), ERK2 (MAPK1), ARRB1, ARRB2, and CREB1 [30,35,45,46,47,49]. Moreover, effective reduction was discernible at the protein level (Figure 2b). A key function of SCF is the maintenance of MC survival. We studied whether RHEX regulates MC viability by applying the YoPro/PI technique, which is well suited to detect even small nuances in skin MC survival [30,35,37,45,49,61]. Gating on living versus apoptotic cells, we unexpectedly found that RHEX inhibits rather than promotes MC survival (Figure 2c,d). A low concentration of SCF (10 ng/mL, provided on two consecutive days during transfection) was used to maintain decent levels of survival. While the percentage of living cells after application of the siRNA protocol was culture- (and therefore donor-) dependent in our primary skin-derived MCs, attenuation of RHEX supported the proportion of viable cells in every single experiment (Figure 2c). Two examples are depicted in Figure 2d. As an additional readout, we assessed the activity of caspase-3, which is detectable early during the course of skin MC apoptosis (preceding YoPro positivity by 1–2 d) [37]. In both cultures employed, caspase-3 activity was reduced by RHEX-siRNA after 24 h (Appendix A), thereby agreeing with the subsequent increase in viable cells by flow-cytometric determination upon RHEX knockdown (Figure 2c,d). We therefore conclude that RHEX counteracts the survival of SCF-supported MCs.

### 3.3. RHEX Is a Negative Regulator of KIT Signaling Restricting Activation of p38 and ERK, but Not of STAT5 or AKT

RHEX was described as a positive regulator of EPO/EPOR signaling [32]. We therefore expected RHEX to act as a positive regulator of KIT-transduced signaling, so that its knockdown would reduce phosphorylation of the major signaling components. Surprisingly, yet in accordance with the above findings for survival, the opposite was true. More precisely, while the level of phospho-AKT and phospho-STAT5 remained unchanged in RHEX-targeted cells, SCF-triggered phosphorylation was enhanced at the level of ERK1/2 and p38 (Figure 3a,b). No difference was found for the signaling intermediates at baseline, whose phosphorylation was undetectable in most cases except for a weak signal for pp38 (and/or pERK2), which occasionally appeared. To ascertain that the detected effects were not a reflection of altered kinetics instead of an overall increase, we repeated the knockdown experiments with larger cell numbers and stimulated for 5, 10, and 15 min in parallel; the increase in pERK1/2 and pp38 in RHEX-targeted MCs was, however, reproducible at different time points (Appendix A).

### 3.4. Comparison between Capicua (CIC) and RHEX: Similar Inhibition of ERK by Both Proteins, but Striking Difference at STAT5

We recently uncovered CIC as a novel repressor of KIT signaling [30]. After revealing RHEX as another repressive component, we compared the impact exerted by RHEX and CIC side by side. CIC was reportedly downregulated after SCF-mediated activation of MCs [30], but RHEX expression remained unaffected by SCF (Appendix A). While attenuation of ERK activation was similar for both proteins, and thus the negative effect lifted upon their knockdown, the lacking effect of RHEX on STAT5 phosphorylation was striking (Figure 4). This finding contrasted with CIC, which had the most prominent impact precisely on STAT5 (Figure 4), in accordance with our recent report [30]. AKT phosphorylation was unaffected by either, and there was likewise no effect on baseline phosphorylation (Figure 4) in accordance with Figure 4 and our previous report [30]. It may therefore be postulated that the two proteins exert suppression by distinct mechanisms and through different (direct or indirect) interactions with KIT. The results also demonstrate that distinct modules downstream of KIT can be regulated independently and thereby uncoupled from each other. A semi-quantification of the two experiments can be found in Appendix A.

### 3.5. RHEX Attenuates SCF-Triggered Expression of Immediate-Early Genes (IEGs)

We recently reported that SCF strongly induces the immediate-early genes NR4A2, JUNB, EGR1, and FOS in skin MCs [30,49]. These genes are barely detectable prior to stimulation but induced sometimes over 1000-fold following KIT ligation in accordance with previous findings for other types of MCs/stimuli combinations [62,63,64,65,66,67,68,69,70,71,72,73,74]. In skin MCs, their induction is entirely ERK-dependent (but PI3K-independent). As a readout of ERK activity, we finally asked whether RHEX attenuation modulates IEG induction. In fact, NR4A2, JUNB, and EGR1 experienced an additional boost when RHEX expression was attenuated (Figure 5a–c), harmonizing with RHEX’s suppressive effect at ERK phosphorylation. Interestingly, FOS behaved differently (Figure 5d), suggesting a bipotential role of RHEX at the FOS promoter. Notwithstanding, we conclude that RHEX represses KIT function in part by inhibiting ERK1/2.

## 4. Discussion

RHEX function is poorly understood. A previous publication reported on a supportive role in erythropoiesis through promotion of EPOR signaling [32]. Information on other lineages, including MCs, is lacking, and there are as yet few entries on PubMed (https://pubmed.ncbi.nlm.nih.gov/) for either RHEX or c1orf186 (last accessed on 25 February 2023).

Thus, this work has led to a number of interesting findings. First, by using publicly available sources, we found strong enrichment or almost exclusive expression of RHEX in the MC lineage at the mRNA and protein levels. Second, the use of MCs encompassing distinct stages of differentiation [55] revealed robust expression already at an early stage, though expression was most pronounced in fully differentiated skin MCs. RHEX expression was also pronounced in hCBMCs in datasets from two distinct laboratories [57,58]. Third, through knockdown experiments in mature skin MCs, we identified RHEX as a repressor of KIT-assisted programs, since SCF-maintained viability was increased upon RHEX knockdown. Fourth, RHEX interfered with SCF-triggered MAPK activation as well as IEG induction, the latter critically relying on ERK1/2. Fifth, by direct comparison with the newly uncovered repressor CIC, we could also demonstrate differential mechanisms of operation between the two proteins. Therefore, in contrast with EPO-supported erythropoiesis, in which RHEX assumes the role of a positive regulator [32], the same protein adopts an inhibitory function in SCF-supported MCs.

At an early stage of development, MC and erythrocytic developmental trajectories overlap, as bi- or tripotential progenitors (the latter also encompassing the megakaryocytic lineage), and vicinity in hematopoietic landscape models have been uncovered, e.g., by the use of single-cell transcriptomics [75,76,77,78,79]. In accordance, the lineages share cell surface receptors, including KIT itself, and transcription factors such as GATA1, SCL/TAL1, GFI1B, and FOG1 [51,60,75,80,81]. It has been proposed that the vicinity of the erythroid and MC lineages may be of advantage during parasitic infections where MC activity leads to helminth expulsion, while there is blood loss along the way that needs to be compensated for. This may have provided the evolutionary pressure that shaped the developmental relationship between these lineages [77].

MCs can even express EPOR [60,82,83], though EPO stimulation does not activate the JAK–STAT pathway in MCs, and rather acts in a complex with KIT to downregulate inflammatory pathways [83].

Given the above overlaps in the molecular repertoires driving mastopoiesis and erythropoiesis, it is appealing to speculate that RHEX serves in the erythrocytic-MC bifurcation by supporting erythrocytes over alternative fates such as MCs. It is of interest in this regard that, in UT7epo, the major cell that has served to delineate aspects of RHEX biology, only EPO was able to stimulate RHEX phosphorylation, while other growth factors, including SCF, did not lead to this posttranslational modification [83]. The variance of KIT signaling independence of the cell subset is underlined by the observation that, in contrast to MCs, STAT5 is not activated by SCF in erythroid cells, while protein kinase A (PKA) is only activated in the latter [30,84]. Another major difference between erythropoiesis and mastopoiesis is that the former requires SCF/KIT and EPO/EPOR in a timely shifted manner, also involving complex interactions between the two receptor systems [85,86,87,88,89,90]. In this regard, the SCF/KIT pair has important functions in early erythropoiesis but may interfere with terminal differentiation of erythroblasts and rather promote less mature precursors and colony-forming units [91]. Conversely, in MCs, SCF/KIT seems to be the dominant system regulating all aspects of mast cell differentiation and function from early precursors to fully differentiated subsets [see Introduction].

Through our phosphoproteomic screen, we also uncovered CIC as a novel inhibitor of the SCF/KIT axis in skin MCs [30]. CIC acts as a transcriptional repressor and is involved in developmental processes; it is also deregulated in cancer [92,93,94,95]. CIC was substantially modified and degraded following SCF in skin MCs and acted as a KIT repressor [30]. Here, we compared CIC and RHEX side by side. While CIC interfered with the activation of ERK1/2, and more potently STAT5, RHEX had no effect on STAT5 phosphorylation, but operated as a repressor of the MAPKs p38 and ERK1/2. RHEX contains putative Grb2 (growth factor receptor-bound protein 2) binding sites, and a physical interaction between Grb2 and RHEX could indeed be established in erythroblasts [32]. Since Grb2 is essential for the activation of the MEK/ERK cascade downstream of KIT, it is conceivable to assume that RHEX may act by sponging Grb2 away from KIT. In favor of this assumption is the fact that Grb2 activation requires Y703 and Y936 on KIT’s intracellular portion [96], while other signaling modules (e.g., PI3K/AKT or STAT5) depend on distinct tyrosine residues in the intracellular tail. RHEX/Grb2 interactions in MCs may thus tentatively explain why MAPKs (but apparently no other modules) are affected by RHEX knockdown. Thus, while RHEX may interfere with the interaction between KIT and Grb2, Grb2-RHEX may further strengthen its binding to EPOR. Future studies will be required to examine this theory in the different lineages side by side. Nonetheless, since RHEX countered the activation of ERK in SCF-stimulated MCs, it accordingly limited the induction of IEGs, genes rapidly induced following stimulation without the necessity of new protein synthesis [97,98]. Therefore, interference with RHEX allowed a more efficient transcription of JUNB, NR4A2, and EGR1. The best-understood target of ERK in the context of IEG induction is ELK1 (ETS-Like Gene 1), a TCF (ternary complex factor) family co-factor of SRF (serum response factor) [97,98]. We recently found that in MCs, ERK-driven IEG expression also strongly depends on CREB (cAMP responsive element binding protein) [49]. Strikingly, RHEX reduction did not lead to greater induction of FOS, however. Even if looking at the data of individual cultures one by one, no tendency could be detected. The altered behavior of FOS may be due to a counteractive process, i.e., another event modified by RHEX in an inverse manner. Notwithstanding, and despite some fine-tuning effects to the opposite, RHEX counters the induction of IEGs in skin MCs by interfering with ERK activation. However, while there is already a reasonable amount of literature available on CIC function in several cell types and in vivo [92,93,94,95] research into RHEX function is in its infancy, another difference is that even though CIC expression is abundant in MCs, the protein is expressed in various lineages across the body. In contrast, RHEX expression is almost exclusively found in MCs.

Collectively, RHEX is expressed at remarkable levels in MCs. We demonstrate herein that RHEX interferes with selected aspects of KIT function. It is plausible to assume that the protein regulates other processes in the lineage as well. While its exploration in vivo using rodents is impossible due to the later emergence of the gene in evolution, MCs will be excellently suited to comprehensively delineate the function of this poorly understood protein in future studies.

## 5. Conclusions 

RHEX expression is strongly enriched in MCs, where upregulation seems to occur during differentiation, while weak expression is found only in a few other cells. RHEX function is almost completely unknown. We show herein that, despite its increase during MC development, RHEX acts as a repressor of KIT-stimulated programs countering SCF-sustained survival, MAPK activation, and subsequent IEG induction. This contrasts with its reported function in erythropoiesis, where it positively contributes to EPO-stimulated programs [32]. We speculate that RHEX may function in lineage specification at the level of a bipotent mast cell/erythrocytic progenitor by supporting erythropoiesis and simultaneously dampening mastopoiesis. By virtue of their vast expression of RHEX, MCs are excellently suited cells to further explore functional implications of RHEX across stimuli and molecular programs. Our study is an important first contribution to this field.

## Figures and Tables

**Figure 2 cells-12-01306-f002:**
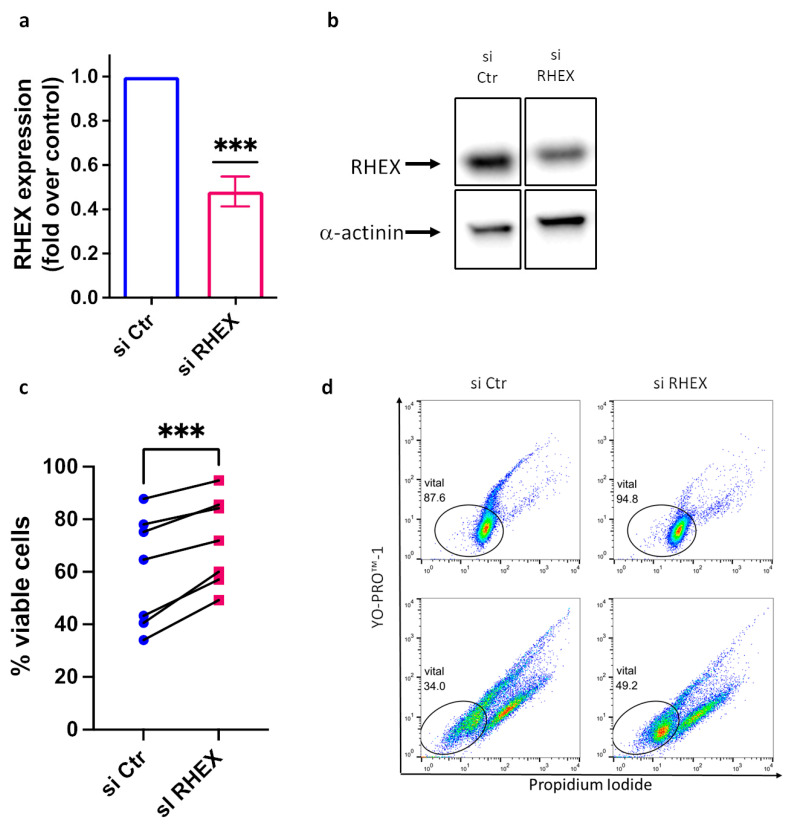
RHEX interferes with SCF-supported survival of skin MCs. (**a**) Skin MCs were treated with control and RHEX-targeting siRNA (in the presence of SCF at 10 ng/mL), and the knockdown efficiency was determined by RT-qPCR after 2 d; it is given as mean ± SEM of n = 7 experiments. (**b**) RHEX knockdown at the protein level. One immunoblot (of two with comparable outcome) is shown; α-actinin served as the loading control. (**c**,**d**) Skin MCs were treated with control and selective siRNAs for 3 d (in the presence of SCF at 10 ng/mL provided on two consecutive days). Viable versus non-viable/apoptotic cells were quantified with the YoPro/PI technique. (**c**) Cumulative results of 7 independent experiments (given as connected dots) show the influence of RHEX expression on the proportion of viable cells. (**d**) Dot plots of two separate experiments, one with low, the other with high overall survival, are depicted (the two cultures correspond to the two extremes in c). *** *p* < 0.001.

**Figure 3 cells-12-01306-f003:**
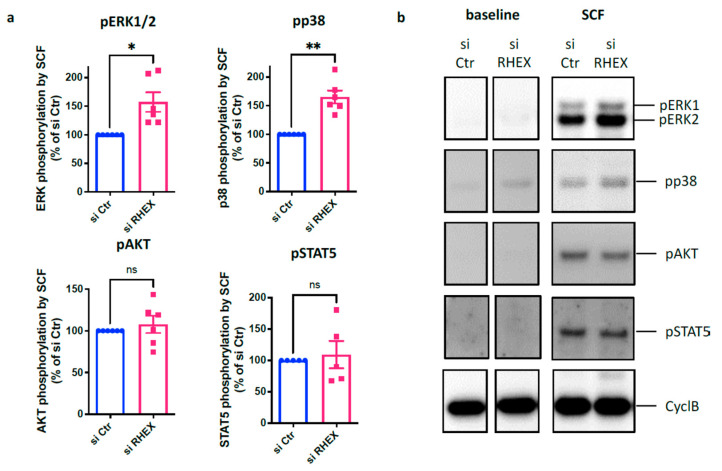
RHEX interferes with SCF-triggered activation of MAPKs (mitogen-activated protein kinases). RHEX silencing was achieved by exposing skin MCs to RHEX-siRNA (si RHEX) for 2 d against control (si Ctr), exactly as in Figure 2a,b; upon silencing, cells were stimulated with SCF (100 ng/mL) for 10 min, signaling components were detected by immunoblotting and semi-quantified by ImageJ. (**a**) cumulative data of 5–6 independent experiments (highlighted by individual dots). (**b**) one representative blot of (**a**) (consecutive detection of the distinct proteins on the same membrane). A time-course experiment at 5, 10, and 15 min following SCF addition is presented in Appendix A. * *p* < 0.05, ** *p* < 0.01, ns – not significant.

**Figure 4 cells-12-01306-f004:**
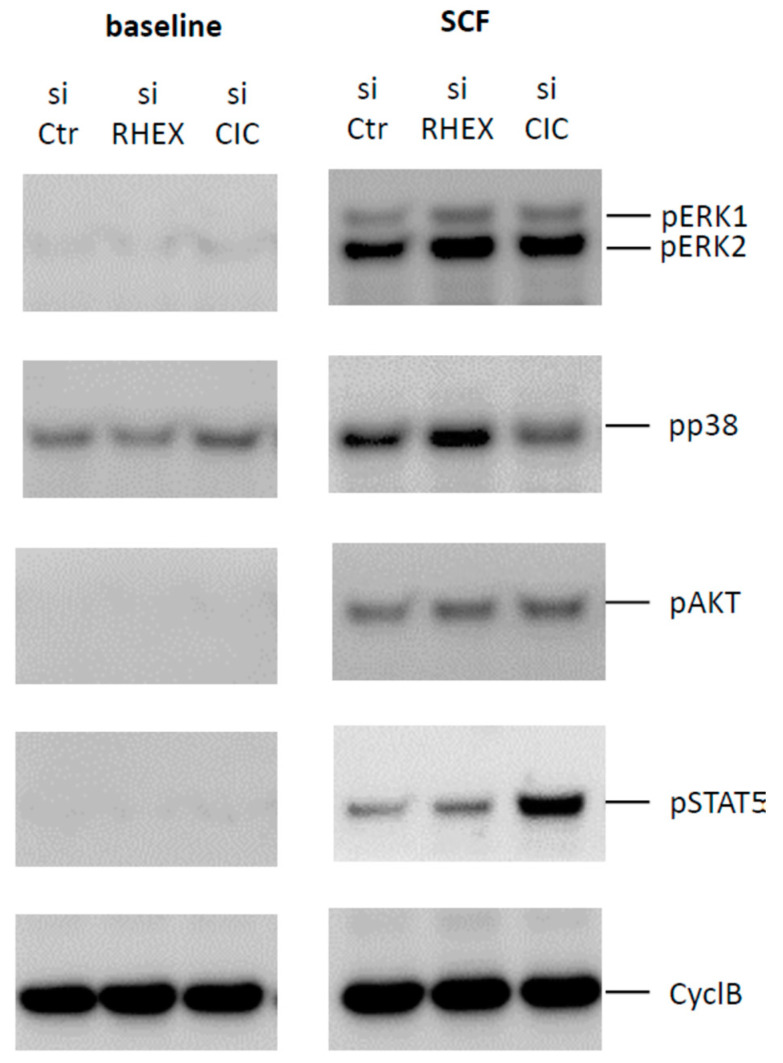
RHEX is a negative regulator of SCF-triggered activation of MAPKs (mitogen-activated protein kinases). RHEX and CIC silencing were achieved by exposing skin MCs to RHEX-siRNA (si RHEX) for 2 d against control (si Ctr), as in Figure 2 and Figure 3. Upon silencing, cells were stimulated with SCF (100 ng/mL) for 10 min or kept without stimulus (baseline). Signaling components were detected by immunoblotting. One representative blot of 2 experiments with comparable outcome is depicted, with consecutive detection of the distinct proteins on the same membrane. CyclB = cyclophilin B was used as the loading control.

**Figure 5 cells-12-01306-f005:**
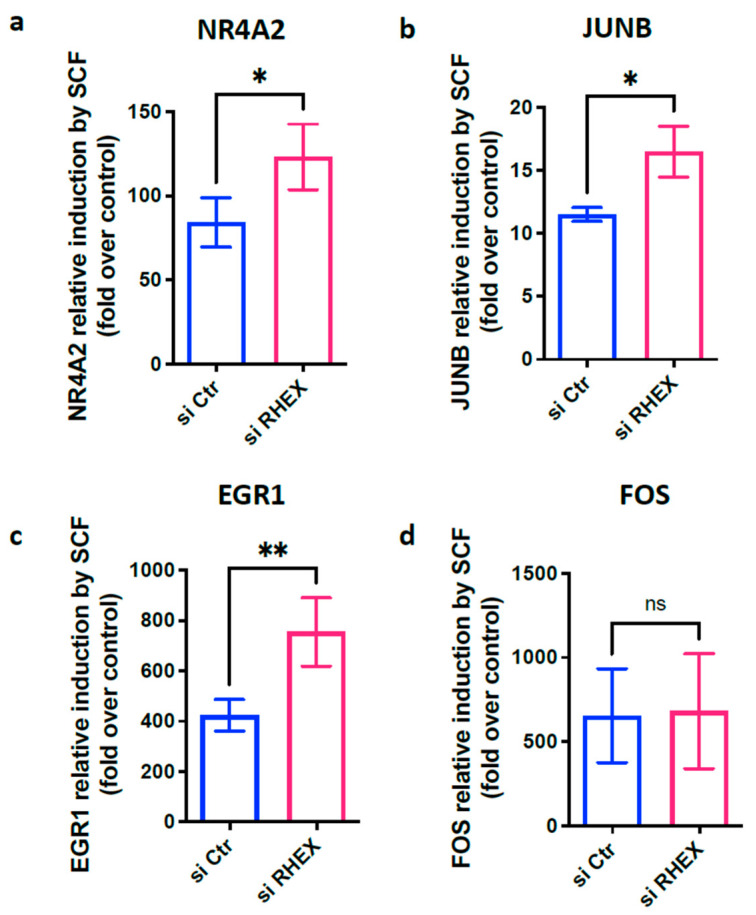
RHEX interferes with the induction of immediate-early gene except for FOS. RHEX silencing was achieved by exposing skin MCs to RHEX-siRNA (si RHEX) versus control siRNA (Ctr si) for 2 d, as in Figure 2 and Figure 3, then stimulated (or not) with SCF for 25 min; RT-qPCR was used to quantitate gene expression (normalized to several housekeeping genes, as described in the Methods). The fold induction by SCF over control is depicted for (**a**) NR4A2, (**b**) JUNB, (**c**) EGR1, and (**d**) FOS. Mean ± SEM of 6–7 experiments (separate cultures). * *p* < 0.05; ** *p* < 0.01; ns—not significant.

**Table 1 cells-12-01306-t001:** Primer pairs used for RT-PCR.

Gene	Forward 5′-3′	Reverse 5′-3′
FOS	AGTGACCGTGGGAATGAAGT	GCTTCAACGCAGACTACGAG
NR4A2	TTCTGTAACCCTCCTAGCCC	AGCATGGCCAAACATTTCCC
JUNB	GCCCGGATGTGCACTAAAAT	GACCAGAAAAGTAGCTGCCG
EGR1	GCCCGGATGTGCACTAAAAT	GACCAGAAAAGTAGCTGCCG
RHEX	TACTGAGAGACGAGGTGCCA	AGTTGATGGCGGTGAGGAAG
HPRT	GCCTCCCATCTCCTTCATCA	CCTGGCGTCGTGATTAGTGA
PPIB *	AAGATGTCCCTGTGCCCTAC	ATGGCAAGCATGTGGTGTTT
GAPDH	ATCTCGCTCCTGGAAGATGG	AGGTCGGAGTCAACGGATTT

***** The PPIB gene encodes Cyclophilin B.

## Data Availability

No datasets were generated during this study. The datasets analyzed in Figure 1 and the text were from public sources [51,52,53,57,58].

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
