# Peer review of "Clorfl86/RHEX Is a Negative Regulator of SCF/KIT Signaling in Human Skin Mast Cells"

_cells, 2023, doi:10.3390/cells12091306_

Round 1

Reviewer 1 Report

In this manuscript, Franke et al examined the regulatory roles of RHEX in human skin mast cells in response to SCF stimulation. They demonstrated that RHEX worked as a negative regulator of SCF-mediated skin mast cell survival by interfering with the Kit signal transduction. This is an interesting study and a better understanding of the signaling pathways downstream of Kit may help elucidate the working mechanisms of mast cells, an important cell type that has been implicated in a wide range of biological processes in addition to its classically defined role in allergy.

Major concerns:

1.     Does treatment of mast cells regulate the expression of RHEX? Given that siRNA roughly reduces the expression of RHEX by 50%, if SCF treatment will differentially upregulate the expression of this protein in the presence or absence of specific siRNA, the assessment of the relevant signaling molecules before and after SCF treatment in specific siRNA-treated versus control siRNA-treated cells may be less precise.  

2.     Under what SCF concentration were the mast cells cultured? If the concentration was similar to the SCF concentration in the experiments shown in Fig 1f, the times (8 min or 20 min) of SCF treatment do not make sense. 

3.     Cord blood-derived mast cells are widely used for investigating the signaling pathways after human mast cell activation. Do CBCMC express RHEX? 

Minor concerns 

1.     Line 208: “Expression remained largely constant in MCs stimulated by FceRI crosslinking (Figure 1a)”. It is not clear whether the cells were activated by FceRI crosslinking according to the figure legend. 

2.     Line 310: Fig 4a and b do not match the text description here. 

3.     Line 332: Fig 5 does not match the text description here. 

4.     Fig 4: Densitometric quantification of the band intensities should be provided. 

5.     Figs 3 and 4: What type of mast cells were used for generating the data in these two figures? 

6.     Line 442: The word “elimination” is not precise here as siRNA treatment only reduces the expression of the protein by 50%, far from elimination.

Reviewer 2 Report

The authors studied the function of RHEX in human mast cells and demonstrated that RHEX was a negative regulator of KIT signal.  The paper is well written, and suitable for publication if the authors consider following points;

1) In Fig. 2d, the dot plots were properly compensated? If possible, gating process and criteria for quadrant analysis should be shown for better understanding.

2) In the point of view of immunology, does RHEX affect the immunological function such as cytokine production of mast cells?

Reviewer 3 Report

The study by Franke et al. demonstrated that RHEX expression is enriched in human mast cells and suppresses SCF/KIT signaling, the RHEX diminution leads to the downstream upregulation of NR4A2, JUNB, and EGR1 induction indicating a selective role of RHEX in human mast cells. The study is overall intriguing, however, several points need to be addressed to ensure the conclusions made by the authors are robust.

1) As the authors mentioned in line 281, they claimed RHEX inhibits mast cell viability through apoptosis, to be accurate, the authors should provide enough evidence to support the claim other than PI staining, such as cleaved caspase-3 staining, etc.

2) The labeling of Figure 2d is very blurry and the percentage of gated cells should be listed more clearly.
